# AN INTERPRETABLE ERROR CORRECTION METHOD FOR ENHANCING CODE-TO-CODE TRANSLATION

**Min Xue**[*]   **Artur Andrzejak**   **Marla Leuther**
Heidelberg University, Germany
`{min.xue, artur.andrzejak, Leuther}@uni-heidelberg.de`

## ABSTRACT

Transformer-based machine translation models currently dominate the field of model-based program translation. However, these models fail to provide interpretative support for the generated program translations. Moreover, researchers frequently invest substantial time and computational resources in retraining models, yet the improvement in translation accuracy is quite limited. To address these issues, we introduce a novel approach, $k$NN-ECD, which combines $k$-nearest-neighbor search with a key-value error correction datastore to overwrite the wrong translations of TransCoder-ST (Roziere et al., 2022). This provides a decision-making basis for interpreting the corrected translations. Building upon this, we further propose $k$NN-ECS$_m$, a methodology that employs a distributed structure with $m$ sub-datastores connected in series, utilizing $m$ diverse experts for multi-round error correction. Additionally, we put forward a *unified name rule*, encouraging the datastore to focus more on code logic and structure rather than diverse rare identifiers. Our experimental results show that our approach improves the translation accuracy from 68.9% to 89.9% of TransCoder-ST (for translation from Java to Python). This error correction method augments program translation, overcoming the inherent limitations of Transformer-based code translation models, such as resource-intensive retraining requirements and uninterpretable outcomes.

## 1 INTRODUCTION

Transformer-based large language models (LLMs) (Vaswani et al., 2017), such as BERT (Devlin et al., 2018), XLM (Lample & Conneau, 2019) and XLNet (Yang et al., 2019), have been widely used in the field of program translation. In recent work, researchers have attempted to enhance Transformer-based program translation by incorporating compiler intermediate representations, but the improvements remain limited (Szafraniec et al., 2023; Rozière et al., 2021). These methods leverage large amounts of data collected from public repositories such as GitHub and GitLab, combining unsupervised and self-supervised learning to overcome the need for parallel corpora. Nevertheless, these techniques often require substantial investments in computational resources for retraining translation models, while yielding only marginal improvements in return. Compared to retraining the LLMs, **Error Correction** emerges as a more efficient alternative, enhancing accuracy through repairing wrong translations produced by the program translation model. This approach holds significant practical value, making it possible to enhance program translation accuracy at a low cost.

The internal workings of Transformer-based models are relatively complex, making it challenging to track and identify which snippets in the training dataset contribute to each output token. In other words, the Transformer-based program translation model cannot provide an intuitive interpretation for the output. A common approach is to exploit knowledge neurons and causal effects to explain the output of the transformer model (Dai et al., 2022; Vig et al., 2020). In the recent research, Meng et al. (2023) revealed the crucial role of middle-layer feed-forward modules in storing factual associations, but it has been so far applied only to simple subject-predicate-object sentences. In contrast, the $k$-nearest-neighbor machine translation ($k$NN-MT) combined with a large-scale datastore has demonstrated a remarkable capacity in optimizing and interpreting natural language translations (Khandelwal et al., 2020; Zhang et al., 2018; Tu et al., 2017). Compared with traditional

---

[*]Corresponding author.

Transformer-based models, datastore possesses inherent interpretability. The $k$NN retrieval method can provide a clear inference path and decision-making basis through tracking and identifying which snippet in the training dataset contribute to each generated token, without the need to analyze complex neural network hierarchies. Based on this, we consider integrating the $k$NN-MT method with error correction, which exhibits great potential in providing an interpretable correction analysis.

In modern source code datasets, among millions of unique identifiers, only less than 1% of the identifiers appear frequently (Karampatsis et al., 2020). Diverse rare identifiers, such as function names, variable names, and parameter names, often increase perturbations during the model training and inference phases (Chirkova & Troshin, 2020b). This phenomenon is akin to introducing noise into the process of information transmission, making it difficult to focus on the key information. Notably, although these different rare identifiers tend to introduce noise in program translation and error correction, only a few researchers have paid attention to this issue. For this problem, we propose a unified name rule to replace diverse rare identifiers during the training and testing phases , minimizing the emphasis on rare identifiers and focusing more on code logic and structure (as shown in Figure 2).

Our work builds upon the Transformer-based code translation models proposed in the TransCoder (Roziere et al., 2020) and TransCoder-ST (Roziere et al., 2022) projects. TransCoder employed self-supervised learning across multiple programming languages (between Java, C++, and Python), and then TransCoder-ST extended it by introducing a rigorously tested parallel corpus. In this paper, we propose to extract error correction information from TransCoder-ST to guide the error correction model in learning repair knowledge. More specifically, we create unit tests for the Java source dataset, and then use TransCoder-ST to generate multiple Python functions with unit tests for each Java function. Subsequently, we conduct unit testing on the Python functions, and then extract the first failed Python function and the first successful Python function to form an error correction language pair for each Java function. Based on this, we establish two alternative error correction models, $k$NN-ECD and $k$NN-ECS$_m$, to improve the translation accuracy of TransCoder-ST by correcting wrong translations. Overall, our contributions are as follows:

- We propose $k$NN-ECD, which leverages $k$NN search to retrieve correction information from the error correction datastore, thereby repairing wrong translations in TransCoder-ST. Notably, the error correction process is interpretable, overcoming the opaque defects of Transformer-based program translation.

- We introduce $k$NN-ECS$_m$, an approach that employs a distributed structure with $m$ small datastore units, capturing comprehensive repair information by using multiple datastore variants for multi-round error correction. This approach promotes the traditional $k$NN-MT technique, effectively improving the data retrieval capabilities by trying diverse data flows.

- We introduce a unified name rule, which ignores the differences in rare identifiers and focuses more on the logic and structure of the code, thus avoiding interference caused by irrelevant identifiers on similarity retrieval.

- We evaluate our approach, showing that the translation accuracy of TransCoder-ST (Java→Python) significantly improves from 68.9% to 89.9% after employing $k$NN-ECD/ $k$NN-ECS$_m$. Without the necessity of retraining Transformer-based models, our approach still achieves substantial improvements in program translation.

## 2    RELATED WORK

**Transformer-based Program Translation.** In recent years, Transformer-based unsupervised learning methods have become the mainstream approach in the field of program translation (Shi et al., 2022; Zhang et al., 2023). TransCoder (Roziere et al., 2020), as the first work that combined an unsupervised model with programming translation, pioneered automatic program translation in the field of software development. Benefitting from the structural characteristics of programming languages, Rozière et al. (2021) introduced a new pre-training objective, DOBF, to recover the original version of the obfuscated source code by pre-training a model. Based on this, Roziere et al. (2022) developed TransCoder-ST, which utilizes an automated unit testing system to filter out invalid translations, and then uses a well-tested parallel corpus to fine-tune the unsupervised model (Radford et al., 2018; Yang et al., 2019; Raffel et al., 2019; Pan et al., 2023). Subsequently, Szafraniec et al.

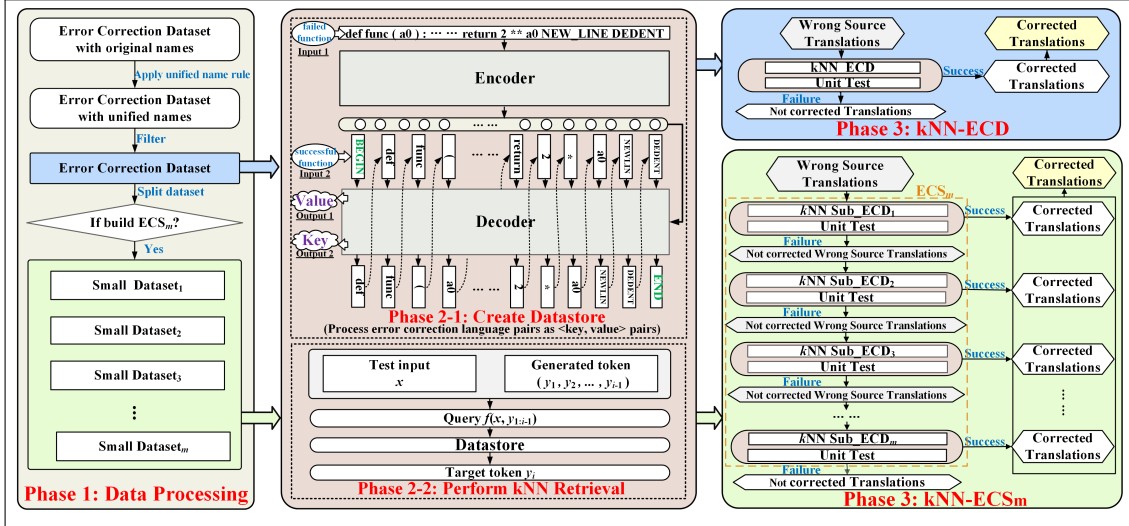

Figure 1: **Training process: Construction workflow of $k$NN-ECD and $k$NN-ECS$_m$.** The workflow consists of three phases. (1) After deduplication, we can get the error correction dataset. If we intend to build $k$NN-ECD, we use the error correction dataset directly; if we intend to build $k$NN-ECS$_m$, we divide the error correction dataset into $m$ small datasets. (2) For the error correction dataset/small_dataset$_{i \in [1,m]}$, we process it into corresponding datastore and introduce the $k$NN retrieval. Here, we can get the corresponding $k$NN-ECD/$k$NN-sub_ECD$_{i \in [1,m]}$, respectively. (3) For $k$NN-ECD, we directly conduct unit testing on the output; for $k$NN-ECS$_m$, we connect sub_ECD$_1, \ldots,$ sub_ECD$_i, \ldots,$ sub_ECD$_m$ in series and carry out unit testing on the output of each sub_ECD$_{i \in [1,m]}$.

(2023) proposed TransCoder-IR, which leverages lower-level compiler intermediate representations to advance code translation. Due to the success of pre-trained language models, Feng et al. (2020) introduced CodeBERT, a neural architecture combining BERT (Devlin et al., 2018), RoBERTa (Liu et al., 2019) and a bidirectional Transformer (Vaswani et al., 2023), trained with a hybrid objective function that incorporates both natural language and programming language. Further, in order to accommodate multilingual representations, Ahmad et al. (2021) employed bidirectional and autoregressive transformer (Lewis et al., 2019) for pre-training on unlabeled natural language and programming language data. Currently, almost all program translation models are based on retraining the Transformer model. However, the complex internal workings of Transformer models leads to uninterpretable translations. Furthermore, this process usually consumes a significant amount of computational resources, yet the gains in translation accuracy are quite limited.

$k$**NN-MT Method.** As a method for providing interpretable results, the datastore-based $k$NN retrieval approach is a promising alternative to Transformer-based translation models. Khandelwal et al. (2020) pioneered the concept of $k$NN-MT, which augments natural language translation by retrieving (key, value) pairs from an external datastore without updating the model. However, large-scale datastores often suffer from high-latency data retrieval. To address this problem, Wang et al. (2021b) introduced a hierarchical clustering strategy (Kanungo et al., 2002), aiming to improve retrieval efficiency by approximately querying the distance between data points in a datastore. On this basis, Wang et al. (2022) proposed to use the cluster-based compact network and cluster-based pruning solution to compress a datastore, thereby reducing the $k$NN retrieval latency. Based on the traditional $k$NN-MT method, researchers have initiated the exploration of more adaptive $k$NN-MT paradigms. Zheng et al. (2021) introduced the concept of adaptive $k$NN-MT, dynamically customizing the number of nearest neighbors for each target token. Furthermore, in order to achieve a more interactive and efficient learning method, Wang et al. (2021a) proposed the $k$NN-over-$k$NN (KoK) method as a plug-and-play solution for online learning combined with human feedback. However, existing works only focus on the retrieval speed and context adaptation while overlooking the limited search capability, failing to capture global information in a large datastore space.

**Diverse Rare Identifiers.** Various rare identifiers often introduce noise in program translation, distracting attention from critical information such as code structure and logic. Modern source code

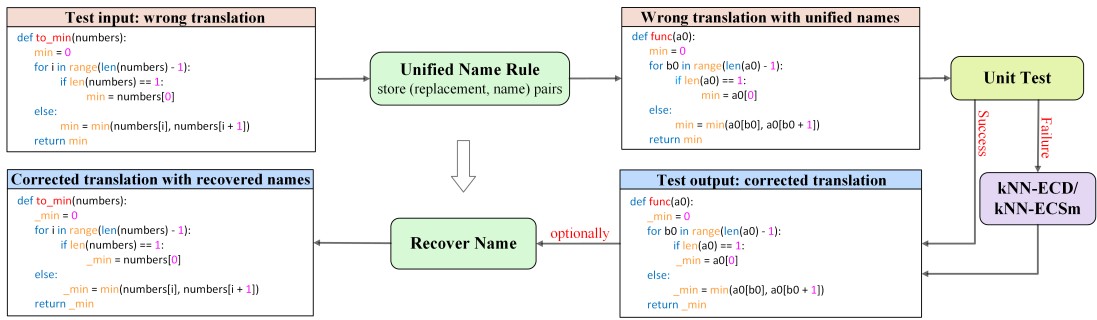

Figure 2: **Testing process: Implementation workflow for $k$NN-ECD and $k$NN-ECS$_m$.** First, we apply the unified name rule on wrong translations and store (replacement, name) pairs. Subsequently, we perform the unit testing on the processed wrong translations, and then feed the failed translations to the error correction model. If necessary, we can recover the masked names in the corrected translations using the (replacement, name) pairs.

datasets contain millions of unique identifiers, but less than 1% of them occur more than 5 times, despite using open vocabulary approaches like BPE (Karampatsis et al., 2020). To address this issue, Chirkova & Troshin (2020b) presented an identifier-anonymization-based approach to handle Out-of-Vocabulary identifiers, which significantly improves Transformer's performance in code completion (Svyatkovskiy et al., 2020) and bug fixing (Gupta et al., 2017; Li et al., 2020). Additionally, Xu et al. (2019) also recognized the limitation of Out-of-Vocabulary terms, proposing to replace all class/variable names with corresponding placeholders. For unfamiliar identifiers, Ahmed et al. (2018) proposed the TRACER method, which handles these cases by replacing all identifiers with their recommended abstract types. Furthermore, by incorporating a more versatile value anonymization process, Chirkova & Troshin (2020a) explored the effectiveness of anonymization across a broader range of tasks based on the Transformer architecture. In previous works, researchers have ignored the possibility of unifying diverse identifiers in both training and test datasets. Also, existing methods are extremely simple, often breaking the code structure and neglecting to avoid replacing special identifiers such as built-in functions.

## 3 APPROACH

In this paper, we aim to enhance code translation through error correction, rather than retraining the program translation model. As shown in Figure 1, following data processing, we create the corresponding datastore and perform $k$NN retrieval on two variants of error correction dataset, constructing two alternative error correction models to enhance TransCoder-ST: $k$NN-ECD and $k$NN-ECS$_m$.

### 3.1 CREATION OF ERROR CORRECTION DATASET

**Generating error correction language pairs.** We intend to extract error correction language pairs from TransCoder-ST, guiding $k$NN-ECD and $k$NN-ECS$_m$ in learning correction knowledge. First, we download the Java source dataset from Google BigQuery and process it using the TransCoder-ST preprocessing pipeline. Subsequently, we employ EvoSuite (Dinella et al., 2022) to create high-quality Java unit test cases, and then feed the processed Java functions with the test cases into TransCoder-ST (Java → Python, beam_size = $N$). For each Java function, we can get $N$ Python functions with their corresponding unit test cases, followed by executing the Python unit tests. If both 'success' and 'failure' occur in $N$ test results, we combine the first failed Python function with the first successful Python function to form an error correction language pair.

**Unified name rule.** Inspired by OOV anonymization method (Chirkova & Troshin, 2020b), we propose a unified name rule to standardize the diverse rare identifiers in the training dataset and test dataset, thus reducing noise in the retrieval process. First, we categorize non-built-in identifiers into three groups: variable names, function name, and parameter names, where each function has only one function name. Then, we replace the identifiers initially used in the code with identifiers following a homogeneous schema of naming, as follows:

*Unified name rule*: For non-built-in identifiers in a function, we sequentially replace parameter names with $a_0, a_1, \ldots, a_j$, function name with $func$, and variable names with $b_0, b_1, \ldots, b_j$. During this process, the (replacement, name) pairs of each function are recorded. If needed, the original name can be recovered.

In Figure 2, we give an example of a unified name rule used for identifier replacement. By implementing the unified name rule, we can filter out extraneous information from rare identifiers, thereby channeling attention toward essential code elements such as logic and structure. Besides, this rule facilitates preliminary error correction within the test dataset, repairing errors resulting from identifier confusion in the code, such as the overlap between function names and variable names.

## 3.2 ERROR CORRECTION USING $k$NN-ECD

**Creating error correction datastore (ECD).** The ECD is generated from the error correction dataset, which mainly includes two primary storage components. The first component is designed to store error correction knowledge in the format of (key, value) pairs, where the generation of (key, value) pair follows $k$NN-MT method (Khandelwal et al., 2020). The second component is used to store source functions and target function prefixes $\langle src\_func, tgt\_func\_pref\rangle$, as well as their corresponding ground truth target tokens $tgt\_tok$. More specifically, we use the pre-trained Transformer model as a coding tool, where key $= f(\langle src\_func, tgt\_func\_pref\rangle)$ is the representation of $\langle src\_func, tgt\_func\_pref\rangle$ obtained from the last hidden layer of the decoder, and value $= h(tgt\_tok)$ is the tokenization of $tgt\_tok$ (Ferrando et al., 2022). For each token generated from (key, value) pairs, users can obtain a detailed explanation by accessing the corresponding $\langle src\_func, tgt\_func\_pref\rangle \rightarrow tgt\_tok$, where $\langle src\_func, tgt\_func\_pref\rangle$ serves as the *decision-making basis* for the generated token $tgt\_tok$.

**Performing $k$NN retrieval on ECD.** Combined with the ECD, we utilize $k$NN retrieval to correct the wrong translations of TransCoder-ST. The $k$NN method conducts the similarity search on a large-scale datastore, retrieving relevant (key, value) pairs to generate error correction results. In the code correction process, when generating the next token $y_i$, at each step, we utilize the representation $f(x, \hat{y}_{1:i-1})$ as a *query*, based on the test input $x$. Following this, we retrieve the $k$ nearest neighbors to the query from the error correction datastore, where $\hat{y}$ represents the generated token. Then, we calculate the distance $d()$ between the key and the query as a weight to regularize the probability of the value. On the basis of $k$NN-MT (Khandelwal et al., 2020), we define the probability distribution of the next token as follows:

$$p\left(y_i|x, \hat{y}_{1:i-1}\right) = \sum_{(k_j, v_j) \in N} \mathbb{1}_{y_i = v_j} exp\left(\frac{-d(k_j, f(x, \hat{y}_{1:i-1}))}{T}\right)$$

where $T$ is the temperature, used to prevent overfitting in the retrieval context. When the temperature value $T > 1$, it tends to make the distribution more uniform, preventing deviations towards the most similar retrieval results and ensuring diversity in the retrieved data.

## 3.3 ERROR CORRECTION USING $k$NN-ECS$_m$

**Creating error correction system (ECS$_m$).** As shown in Figure 1, ECS$_m$ consists of $m$ sub-datastores, which are connected in sequential order. To build ECS$_m$, we first randomly divide the error correction dataset into $m$ equal parts {small\_dataset$_1$, …,small\_dataset$_i$,…, small\_dataset$_m$}, where each small\_dataset$_{i \in [1,m]}$ contains the same number of error correction language pairs. It means that ECD and ECS$_m$ are generated from the same error correction dataset. Then, we follow the ECD creation process to generate the corresponding {sub\_ECD$_1$, …, sub\_ECD$_i$, …, sub\_ECD$_m$}, where these sub\_ECDs are linked sequentially. Here, each sub\_ECD$_{i \in [1,m]}$ contains two primary storage components. The first components is dedicated to storing (key, value) pairs, while the second components is designed for storing $\langle src\_func, tgt\_func\_pref\rangle \rightarrow tgt\_tok$ records.

**Performing $k$NN retrieval on ECS$_m$.** Within the framework of ECS$_m$, we repeatedly feed incorrected wrong translations into subsequent sub\_ECD$_{i \in [1,m]}$ and perform $k$NN retrieval, using diverse data flows for multiple rounds of error correction. Specifically, for each sub\_ECD$_{i \in [1,m]}$, we employ $k$NN retrieval to generate outputs and conduct unit testing on the generated results. If the unit testing

result is successful, it means that the wrong translation has been corrected. In this case, we output the corrected code. However, if the unit testing result failed, it means that the wrong translation has not been adequately corrected. In response, we re-enter the wrong source translation into the next sub-datastore. By adopting a distributed structure with multiple sub-datastores, $k$NN retrieval can capture more comprehensive repair information, effectively enhancing the $k$NN search capability.

# 4 EXPERIMENTS

## 4.1 EXPERIMENTAL DETAILS

**Model architecture.** The implementation of $k$NN-ECD/$k$NN-ECS$_m$ is conducted under the guidance of oracle (Loney & McClain, 2004), where oracle represents a group of manual unit testers. By reviewing the output and providing feedback, oracle assists us in determining whether the corrected results make sense. As shown in Figure 2, we implement the unified name rule on the test input for preliminary error correction before feeding it into $k$NN-ECD/$k$NN-ECS$_m$. Then, we execute unit testing on the processed test dataset. If successful, it means that the wrong translation is due to identifier confusion. If failed, we feed the failed test samples into ECD/ECS$_m$. (Here, we define the model as $k$NN-ECD$^*$/$k$NN-ECS$^*_m$ when the unified name rule is not employed during the training and testing phases.)

**Training dataset & test dataset.** We download the Java source code from Google BigQuery and implement the TransCoder-ST preprocessing pipeline for dataset filtering. Next, we set the maximum runtime of 20 seconds for each process and employ EvoSuite to create Java test cases. Here, the unit test cases are created based on two criteria: mutation score over 0.9 and at least two assertions. In this process, we collect 82,665 Java functions with unit test cases.

Subsequently, we feed the above processed Java function into TransCoder-ST (beam_size = 10), and set the target language as Python. In this step, $82,665 \times 10$ Python functions with unit test cases are generated. For each Java function, we perform unit testing on the corresponding 10 Python functions. If 'success' and 'failure' both appear in 10 test results, we merge the first failed Python function and the first successful Python function into an error correction language pair. Ultimately, we can get a preliminary error correction dataset with 26,230 language pairs for TransCoder-ST.

After that, we employ the unified name rule to standardize diverse rare identifiers within the preliminary error correction dataset. By removing duplicate entries from the processed dataset, an error correction dataset with 21,385 correction pairs is built. Then, we divide the error correction dataset into a training dataset and a test dataset with a ratio of $9 : 1$. Among them, the training dataset consists of 19,368 error correction language pairs, and the test dataset consists of 2,017 wrong translations, each with unit test cases and (replacement, name) pairs[1].

**Training details & fine-tuning details.** During the training phase, we introduce the previous version of TransCoder-ST as a coding tool for error correction language pairs, mainly because its encoder and decoder can simultaneously process codes in the same programming language, which conforms to the data properties of error correction language pairs (Qi et al., 2018). Based on this, we feed the error correction language pair into the coding tool, where we treat the correct translation as *target* and the wrong translation as *source*. Then, we extract the cross-attention output as *key*, and the tokenization of the ground truth target token as *value*.

During the fine-tuning phase, we evaluate the performance of ECD and ECS$_m$ with different numbers of sub-datastores, where $m \in \{3, 6, 9, 12, 15, 18\}$. We consider the following parameters: neighbor ($p_0$) and temperature ($p_1$). When more neighbors are retrieved, noise may be introduced, which can lead to worse results. Meanwhile, properly adjusting the temperature parameters can prevent excessive bias towards a single neighbor, thus ensuring the diversity of results. In this case, we test the performance of ECD and ECS$_m$ under $p_0 \in \{1, 4, 8, 12, 16, 32\}$, $p_1 \in \{1, 10, 100, 1000\}$, and select the optimal parameter combination. Experiments show that for ECD, when $p_0, p_1 = \{4, 10\}$, the best error correction rate can be achieved. For ECS$_m$ with $m = 3, 12, 15$, the optimal parameter combination is $p_0, p_1 = \{8, 10\}$. ECS$_m$ with $m = 6, 18$ attains the highest effectiveness with the parameter combination $p_0, p_1 = \{16, 10\}$. As for ECS$_m$ with $m = 9$, the most effective parameter combination is $p_0, p_1 = \{2, 10\}$.

---

[1]https://github.com/minxue29031/Error_Correction

## 4.2 RESULTS AND DISCUSSION

**Translation performance.** In Table 1, we compare the translation performance of TransCoder-ST after combining ECD/ECS$_m$ or ECD$^*$/ECS$_m^*$. In the comparison, we focus on the translation from Java to Python, where the performance of our models is measured against j2py[2], TransCoder, DOBF, TransCoder-ST (TC-ST). As shown in Table 1, it is clear that when TransCoder-ST utilizes the error correction model, the translation performance improves significantly, increasing from 68.9% to a range of 82.4% ∼ 89.9%. This improvement stems from the datastore, which stores a large amount of error correction information generated from TransCoder-ST. By systematically retrieving relevant ⟨key, value⟩ pairs, we can correct relative errors in the wrong translations. Moreover, comparing $k$NN-ECD/$k$NN-ECS$_m$ with $k$NN-ECD$^*$/$k$NN-ECS$_m^*$, the former shows superior performance. This difference is attributed to the unified name rule, which reduces the interference caused by rare identifiers during datastore construction and implementation.

Table 1: Translation performance of TransCoder-ST with $k$NN-ECD/$k$NN-ECS$_m$ (Java → Python).

| | $k$NN-ECS$_{18}$/ $k$NN-ECS$_{18}^*$ | $k$NN-ECS$_{15}$/ $k$NN-ECS$_{15}^*$ | $k$NN-ECS$_{12}$/ $k$NN-ECS$_{12}^*$ | $k$NN-ECS$_9$/ $k$NN-ECS$_9^*$ | $k$NN-ECS$_6$/ $k$NN-ECS$_6^*$ | $k$NN-ECS$_3$/ $k$NN-ECS$_3^*$ | $k$NN-ECD/ $k$NN-ECD$^*$ |
|---|---|---|---|---|---|---|---|
| j2py | 38.3% | 38.3% | 38.3% | 38.3% | 38.3% | 38.3% | 38.3% |
| TransCoder | 49.0% | 49.0% | 49.0% | 49.0% | 49.0% | 49.0% | 49.0% |
| DOBF | 52.7% | 52.7% | 52.7% | 52.7% | 52.7% | 52.7% | 52.7% |
| Pure TC-ST (Online) | 68.9% | 68.9% | 68.9% | 68.9% | 68.9% | 68.9% | 68.9% |
| TC-ST+ECD$^*$/ECS$_m^*$ | 89.4% | 89.1% | 88.7% | 87.9% | 87.1% | 85.1% | 82.4% |
| TC-ST + ECD/ECS$_m$ | **89.9%** | 89.6% | 89.3% | 89.1% | 87.9% | 86.5% | 84.5% |

[*] We cannot directly compare our results with the latest research, TransCoder-IR, because it is not suitable for translating from Java to Python.

**Interpretability of error correction model.** During the testing phase, the traditional Transformer model cannot track and identify which snippets in the training dataset contributed to each generated token. However, $k$NN-ECD/$k$NN-ECS$_m$ can address this issue, providing an intuitive and readable decision-making basis for each output token. When building the datastore, we store the ⟨key, value⟩ pairs and the corresponding snippets ⟨$src\_func, tgt\_func\_pref$⟩ → $tgt\_tok$ from the training dataset, where the coding of ⟨$src\_func, tgt\_func\_pref$⟩ is key, the coding of $tgt\_tok$ is value, and ⟨$src\_func, tgt\_func\_pref$⟩ serves as the decision-making basis for the generated token $tgt\_tok$. Consequently, when generating each output token, we will return both ⟨key, value⟩ pair and ⟨$src\_func, tgt\_func\_pref$⟩ → $tgt\_tok$. In APPENDIX Table 7, we show a detailed decision-making process for generating a corrected translation. For clarity, we only focus on the correction process of the wrong token '/' → '//' in Table 2, where the ⟨key, value⟩ pair used to correct the wrong token '/' → '//' is a string of numbers, making it challenging to extract valid intuitive information. In such cases, the uncoded form ⟨·, ·⟩ → '//' of ⟨key, value⟩ pair can provide more intuitive information.

Table 2: Interpretable error correction process for repairing incorrect tokens in wrong translations

| Wrong translation | Corrected translation | Decision-making basis |
|---|---|---|
| ```def func ( a0 ) :\n    return 9 * a0 / 5 + 32``` | ```def func ( a0 ) :\n    return 9 * a0 // 5 + 32``` | ⟨ 'def func ( a0 ) : NEW_LINE INDENT return a0 / 6 NEW_LINE DEDENT', 'def func ( a0 ) : NEW_LINE INDENT return a0'⟩ → '//' |
| ```def func ( a0 ) :\n    b = list ( a0 )\n    b . sort ( )\n    return b [ len ( b ) / 2 ]``` | ```def func ( a0 ) :\n    b = list ( a0 )\n    b . sort ( )\n    return b [ len ( b ) // 2 ]``` | ⟨ 'def func ( a0 ) : NEW_LINE INDENT a0 . sort ( ) NEW_LINE return a0 [ len ( a0 ) / 2 ] NEW_LINE DEDENT', 'def func ( a0 ) : NEW_LINE INDENT a0 . sort ( ) NEW_LINE return a0 [ len ( a0 )'⟩ → '//' |
| ```def func ( a0 ) :\n    b = 0\n    while a0 > 0 :\n        if a0 % 10 == 2 :\n            b += 1\n        a0 = a0 / 10\n    return b``` | ```def func ( a0 ) :\n    b = 0\n    while a0 > 0 :\n        if a0 % 10 == 2 :\n            b += 1\n        a0 = a0 // 10\n    return b``` | ⟨ 'def func ( a0 ) : NEW_LINE sum = 0 NEW_LINE while a0 > 0 : NEW_LINE INDENT sum += ( a0 % 10 ) ** 2 NEW_LINE a0 = a0 / 10 NEW_LINE DEDENT return sum NEW_LINE DEDENT', 'def func ( a0 ) : NEW_LINE INDENT sum = 0 NEW_LINE while a0 > 0 : NEW_LINE INDENT sum += ( a0 % 10 ) ** 2 NEW_LINE a0 = a0'⟩ → '//' |

[*] ⟨·, ·⟩ → '//' : ⟨·, ·⟩ represents the decision-making basis of the correction process '/' → '//', ⟨·, ·⟩ indicates ⟨$src\_func, tgt\_func\_pref$⟩, and '//' indicates $tgt\_tok$.

**Generalizability analysis.** To explore the generalization of the error correction model, first, we analyze the overlap of code text between the test dataset and the training dataset. Following data preprocessing, we observe that there are no duplicated codes between the two datasets. Additionally, comparing the wrong code fragments (i.e., code fragment containing the wrong token) in the

---

[2]https://github.com/natural/java2python

test dataset with those in the training dataset, we still cannot find any identical wrong code fragments. It means that the $k$NN-ECD/$k$NN-ECS$_m$ can learn how to correct wrong tokens, rather than accidentally correcting wrong tokens due to the test dataset containing duplicate 'codes' or 'wrong code fragments' with the training dataset. Second, we show the interpretable decision-making basis for wrong token fixing in Table 2, where $\langle src\_func, tgt\_func\_pref \rangle \rightarrow tgt\_tok$ is retrieved to fix the wrong token '/' $\rightarrow$ '//'. We find that the text of the wrong translation is quite different from the text of $\langle src\_func, tgt\_func\_pref \rangle$. It indicates that the error correction model can well apply the knowledge learned from the error correction dataset to new samples. In summary, the above error correction behavior demonstrates that the error correction model can extend the repair knowledge acquired from the training dataset to new wrong translations, rather than merely relying on straightforward comparisons of similar code texts.

**Error correction performance of $k$NN-ECD$^*$ and $k$NN-ECS$_m^*$.** In Table 3, we compare the error correction performance of $k$NN-ECD$^*$ and $k$NN-ECS$_m^*$. It is worth noting that $k$NN-ECD$^*$ and $k$NN-ECS$_m^*$ are generated from the same error correction dataset, which means that both contain the same error correction information. Surprisingly, in this case, $k$NN-ECS$_m^*$ achieves 65.8% error correction rate, showing a substantial 22.3% improvement compared to $k$NN-ECD$^*$. Besides, as shown in Figure 4(a), with the increase in the number of sub-datastores under the error correction system, we observe an improvement in error correction performance. The main reason is that traditional $k$NN methods usually suffer from insufficient retrieval capabilities in a large datastore, which tends to focus on high-density regions while failing to capture potential correlations in low-density regions. In contrast to $k$NN-ECD$^*$, the improvement of $k$NN-ECS$_m^*$ is primarily attributed to its distributed structure, which includes diverse datastore variants. By employing different data flows for multiple rounds of error correction, it can capture more comprehensive error correction information.

Table 3: Correction performance of $k$NN-ECD$^*$/$k$NN-ECS$_m^*$ on TransCoder-ST wrong translations

| | $k$NN-ECS$_{18}^*$ | $k$NN-ECS$_{15}^*$ | $k$NN-ECS$_{12}^*$ | $k$NN-ECS$_9^*$ | $k$NN-ECS$_6^*$ | $k$NN-ECS$_3^*$ | $k$NN-ECD$^*$ |
|---|---|---|---|---|---|---|---|
| sub_ECD$_1^*$ | 25.4% | 26.4% | 27.3% | 28.5% | 30.7% | 33.9% | 43.5% |
| sub_ECD$_2^*$ | 26.5% | 27.2% | 28.6% | 30.0% | 33.0% | 36.3% | - |
| sub_ECD$_3^*$ | 26.5% | 27.4% | 28.0% | 28.7% | 32.0% | 35.9% | - |
| sub_ECD$_4^*$ | 26.1% | 27.9% | 30.4% | 29.3% | 30.9% | - | - |
| sub_ECD$_5^*$ | 25.0% | 26.2% | 29.5% | 28.0% | 31.4% | - | - |
| sub_ECD$_6^*$ | 26.0% | 26.6% | 28.6% | 29.5% | 32.1% | - | - |
| sub_ECD$_7^*$ | 25.4% | 29.1% | 28.7% | 28.4% | - | - | - |
| sub_ECD$_8^*$ | 24.1% | 26.7% | 26.8% | 26.9% | - | - | - |
| sub_ECD$_9^*$ | 25.9% | 27.0% | 26.8% | 30.2% | - | - | - |
| sub_ECD$_{19}^*$ | 27.9% | 27.6% | 27.3% | - | - | - | - |
| sub_ECD$_{11}^*$ | 26.9% | 26.8% | 28.5% | - | - | - | - |
| sub_ECD$_{12}^*$ | 25.7% | 24.9% | 27.8% | - | - | - | - |
| sub_ECD$_{13}^*$ | 27.7% | 26.0% | - | - | - | - | - |
| sub_ECD$_{14}^*$ | 27.1% | 27.0% | - | - | - | - | - |
| sub_ECD$_{15}^*$ | 26.0% | 27.6% | - | - | - | - | - |
| sub_ECD$_{16}^*$ | 26.2% | - | - | - | - | - | - |
| sub_ECD$_{17}^*$ | 26.0% | - | - | - | - | - | - |
| sub_ECD$_{18}^*$ | 26.3% | - | - | - | - | - | - |
| sub_ECD$_{avg}^*$ | 26.1% | 27.0% | 28.2% | 28.8% | 31.7% | 35.4% | 43.5% |
| ECD$^*$/ECS$_m^*$ | **65.8%** | 64.9% | 63.8% | 61.1% | 58.4% | 52.1% | 43.5% |

**Impact of the unified name rule.** In Figure 4, we show the error correction performance of $k$NN-ECD and $k$NN-ECS$_m$ after applying the unified name rule. Comparing Table 3 and Table 4, we observe that implementing the unified name rule can effectively enhance the error correction rate, ranging from $1.7\% \sim 6.5\%$. The main reason is that, during the training and testing phases, using the unified name rule can ignore the diversity of function names, parameter names, and variable names as much as possible, while paying more attention to the logic and structure of the code. By adopting the unified name rule, we can minimize the emphasis on rare identifiers, thereby eliminating the interference produced by diverse rare identifiers during the process of datastore construction and implementation. Furthermore, it is worth mentioning that employing the unified name rule alone can achieve 5.1% preliminary error corrections before feeding the wrong translation into ECD/ECS$_m$, where the errors mainly arise from identifier confusion within the code.

**Ablation analysis.** In Table 5, we investigate the importance of adjacent tokens in the process of identifying and correcting wrong tokens within the input. Our approach is to progressively remove the adjacent tokens near the target wrong token in the input. This systematic approach provides a

Table 4: Correction performance of $k$NN-ECD/$k$NN-ECS$_m$ on TransCoder-ST wrong translations

| | $k$NN-ECS$_{18}$ | $k$NN-ECS$_{15}$ | $k$NN-ECS$_{12}$ | $k$NN-ECS$_9$ | $k$NN-ECS$_6$ | $k$NN-ECS$_3$ | $k$NN-ECD |
|---|---|---|---|---|---|---|---|
| sub_ECD$_1$ | 29.0% | 29.4% | 30.4% | 32.1% | 35.5% | 42.5% | 48.7% |
| sub_ECD$_2$ | 26.3% | 27.1% | 29.9% | 32.8% | 37.1% | 37.8% | - |
| sub_ECD$_3$ | 26.0% | 26.1% | 27.3% | 33.8% | 33.1% | 37.1% | - |
| sub_ECD$_4$ | 25.3% | 28.4% | 27.9% | 29.0% | 33.7% | - | - |
| sub_ECD$_5$ | 25.3% | 29.7% | 30.9% | 30.4% | 33.2% | - | - |
| sub_ECD$_6$ | 28.1% | 25.8% | 32.0% | 30.0% | 32.5% | - | - |
| sub_ECD$_7$ | 28.3% | 26.9% | 31.0% | 28.4% | - | - | - |
| sub_ECD$_8$ | 26.3% | 29.9% | 28.0% | 30.9% | - | - | - |
| sub_ECD$_9$ | 25.0% | 29.4% | 29.8% | 29.2% | - | - | - |
| sub_ECD$_{10}$ | 25.7% | 28.5% | 28.1% | - | - | - | - |
| sub_ECD$_{11}$ | 26.7% | 28.0% | 28.1% | - | - | - | - |
| sub_ECD$_{12}$ | 26.5% | 25.8% | 26.7% | - | - | - | - |
| sub_ECD$_{13}$ | 29.1% | 27.1% | - | - | - | - | - |
| sub_ECD$_{14}$ | 27.8% | 29.3% | - | - | - | - | - |
| sub_ECD$_{15}$ | 27.3% | 27.2% | - | - | - | - | - |
| sub_ECD$_{16}$ | 25.2% | - | - | - | - | - | - |
| sub_ECD$_{17}$ | 25.8% | - | - | - | - | - | - |
| sub_ECD$_{18}$ | 28.0% | - | - | - | - | - | - |
| sub_ECD$_{avg}$ | 26.8% | 27.9% | 29.2% | 30.7% | 34.2% | 39.2% | 48.7% |
| Unified name rule | 5.1% | 5.1% | 5.1% | 5.1% | 5.1% | 5.1% | 5.1% |
| ECD/ECS$_m$ | **67.5%** | 66.7% | 65.7% | 64.8% | 61.2% | 56.5% | 50.0% |

deeper understanding of how adjacent tokens contribute to the identification and correction of wrong tokens, depending on whether the wrong token is successfully corrected in the output. In Table 5, the wrong token '/' in the wrong translation should be corrected to '//'. Initially, when we feed the original input to the error correction model, it can effectively correct the wrong token '/'. However, as we systematically remove the adjacent tokens one by one, we encounter challenges in rectifying the wrong token. It means that adjacent tokens are key to identifying and correcting wrong tokens. The identification and correction of wrong tokens relies on the internal relationships among adjacent tokens, rather than directly pinpointing the wrong tokens. In essence, the error correction method corrects wrong translations by learning and understanding the internal relationships between tokens within a function.

Table 5: The impact of adjacent tokens in the error correction process

| | Input (original format) | Input (remove len ( )) | Input (remove b) | Input (remove return) |
|---|---|---|---|---|
| Input | ```def func(a0):\n    b = 0\n    for c in a0:\n        b += c\n    return b / len(a0)``` | ```def func(a0):\n    b = 0\n    for c in a0:\n        b += c\n    return b / a0``` | ```def func(a0):\n    b = 0\n    for c in a0:\n        b += c\n    return / len(a0)``` | ```def func(a0):\n    b = 0\n    for c in a0:\n        b += c\n    b / len(a0)``` |
| Output | ```def func(a0):\n    b = 0\n    for c in a0:\n        b += c\n    return b // len(a0)``` | ```def func(a0):\n    b = 0\n    for c in a0:\n        b += c\n    return b / a0``` | ```def func(a0):\n    b = 0\n    for c in a0:\n        b += c\n    return b / len(a0)``` | ```def func(a0):\n    b = 0\n    for c in a0:\n        b += c\n    b / len(a0)\n    return b``` |

## 5 CONCLUSION

In real-world scenarios, the interpretability of outputs plays a crucial role in gaining users' trust. Currently, Transformer-based models are widely used for program translation. However, even with researchers investing significant time and computational resources in retraining models, the improvement in translation accuracy remains relatively limited. Furthermore, due to the complex internal workflow of the Transformer model, it is difficult to track and identify which snippet in the training dataset contribute to each output token. In this paper, we employ the $k$NN retrieval on an error correction datastore to enhance the translation capability of TransCoder-ST through code correction. This approach provides a decision-making basis for each generated token, laying a solid research foundation for the subsequent improvement of error correction. Importantly, by simply integrating additional error correction datastore, the datastore-based $k$NN retrieval approach significantly enhances the translation performance of TransCoder-ST, without the need to consume significant computational resources to retrain the Transformer-based model.

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

# A APPENDIX

## A.1 EVALUATION

Most program translation studies employ BLEU scores to assess the quality of the results (Papineni et al., 2001; Chen et al., 2018; Aggarwal K, 2015). However, a wrong translation typically involves only a few wrong tokens, which means that the BLEU score tends to remain relatively high, regardless of the correctness of the translation. Inspired by computational accuracy in TransCoder (Roziere et al., 2020), we introduce *functional equivalence* to measure the quality of code correction.

*Functional equivalence*: Given the same input, the source code and the corrected code produce the same output, i.e., both the source code and the corrected code succeed under the same unit test cases.

This metric emphasizes the correctness and functionality of the code, rendering it better suited for evaluating the quality of corrected translations. For example, some wrong translations occur not due to bugs in the code, but because the source code and the translated code fail to produce the same output when given the same input.

## A.2 GENERATION OF PYTHON UNIT TEST DATASET

**Why do we need to use translated Python unit tests?** Firstly, in the process of generating the error correction training dataset, we translate a Java function to multiple Python functions (beam_size > 1), and then combine the first failed Python function with the first successful Python function to form an error correction language pair. Therefore, it is crucial to ensure that the Java function and its corresponding multiple Python functions have equivalent unit test dataset, as well as multiple Python functions under the same Java function have the same unit test dataset. Secondly, during the testing phase, we need to verify that the wrong Python function, the corrected Python function, and its source Java function have an equivalent unit test dataset, as well as the wrong Python function and the corrected Python function have the same unit test dataset. Based on the above two cases, directly translating Java unit tests into Python unit tests is a simple and effective approach.

**How to ensure the feasibility of the translated unit test dataset?** For the translation process[3] from Java unit test dataset to Python unit test dataset and the screening process of source Java functions, we refer to the work of TransCoder-ST (Roziere et al., 2020). In Table 6, we show the process of using the error correction model to rectify the wrong Python function generated by TransCoder-ST, where Python unit tests are the translations of Java unit tests. In our training dataset and test dataset, each wrong Python function has a matching correct function (ground truth) that passes the corresponding unit test dataset, and almost every unit test dataset contains 3∼6 unit tests. This ensures that in the training dataset, the wrong function and the correct function under the same error correction language pair use the same unit test dataset. It also guarantees that during the testing phase, if the wrong function can be corrected, the corrected function will successfully pass all unit tests. More importantly, multiple unit tests under each unit test dataset assure the reliability of the final unit test result.

Table 6: Translation process from Java unit tests to Python unit tests

| Type | Details |
|------|---------|
| Source Java function | ```java
public static double clamp (double value, double min, double max) {
    if (value < min) {
        return min;
    }
    if (value > max) {
        return max;
    }
    return value;
}
``` |

Continued on next page

---

[3]`https://github.com/facebookresearch/CodeGen/blob/main/codegen_sources/test_generation/evosuite_tests_translators/evosuite_to_python.py`

| Type | Details |
|---|---|
| Wrong Python function | <pre>def func (a0, a1, a2):
    if a0 < a1:
        return min
    if a0 > a2:
        return max
    return a0</pre> |
| Corrected Python function | <pre>def func (a0, a1, a2):
    if a0 < a1:
        return a1
    if a0 > a2:
        return a2
    return a0</pre> |
| Java unit test dataset | <pre># This file was automatically generated by EvoSuite
# Thu Jan 26 21:55:01 GMT 2023

import org.junit.Test;
import static org.junit.Assert.*;
import org.evosuite.runtime.EvoRunner;
import org.evosuite.runtime.EvoRunnerParameters;
import org.junit.runner.RunWith;

@RunWith(EvoRunner.class) @EvoRunnerParameters(mockJVMNonDeterminism = true, useVFS =
    true, useVNET = true, resetStaticState = true, separateClassLoader = true)
public class CLASS_64ef580337f1_ESTest extends CLASS_64ef580337f1_ESTest_scaffolding {

  @Test(timeout = 4000)
  public void test0()  throws Throwable  {
      double double0 = CLASS_64ef580337f1.clamp((-1275.777374486698),
          (-72868.4857075), 42848.99);
      assertEquals((-1275.777374486698), double0, 1.0E-4);
  }

  @Test(timeout = 4000)
  public void test1()  throws Throwable  {
      double double0 = CLASS_64ef580337f1.clamp(44640.610466346, (-42604.8), (-1.0));
      assertEquals((-1.0), double0, 1.0E-4);
  }

  @Test(timeout = 4000)
  public void test2()  throws Throwable  {
      double double0 = CLASS_64ef580337f1.clamp((-2306.1415894012503),
          44640.610466346, (-1.0));
      assertEquals(44640.610466346, double0, 1.0E-4);
  }

  @Test(timeout = 4000)
  public void test3()  throws Throwable  {
      double double0 = CLASS_64ef580337f1.clamp(0.0, 0.0, (-1.0));
      assertEquals((-1.0), double0, 1.0E-4);
  }

  @Test(timeout = 4000)
  public void test4()  throws Throwable  {
      double double0 = CLASS_64ef580337f1.clamp((-84626.14348206822), 0.0, 0.0);
      assertEquals(0.0, double0, 1.0E-4);
  }

  @Test(timeout = 4000)
  public void test5()  throws Throwable  {
      CLASS_64ef580337f1 cLASS_64ef580337f1_0 = new CLASS_64ef580337f1();
  }
}</pre> |

| Type | Details |
|---|---|
| Python unit test dataset | |

```python
import numpy as np
import math
from math import *
import collections
from collections import *
import heapq
import itertools
import random
import sys
import unittest

#TOFILL
class CLASS_64ef580337f1(unittest.TestCase):
  def test0(self):
      double0 = f_filled((-1275.777374486698), (-72868.4857075), 42848.99)
      assert abs((-1275.777374486698) - double0) <= 1.0E-4

  def test1(self):
      double0 = f_filled(44640.610466346, (-42604.8), (-1.0))
      assert abs((-1.0) - double0) <= 1.0E-4

  def test2(self):
      double0 = f_filled((-2306.1415894012503), 44640.610466346, (-1.0))
      assert abs(44640.610466346 - double0) <= 1.0E-4

  def test3(self):
      double0 = f_filled(0.0, 0.0, (-1.0))
      assert abs((-1.0) - double0) <= 1.0E-4

  def test4(self):
      double0 = f_filled((-84626.14348206822), 0.0, 0.0)
      assert abs(0.0 - double0) <= 1.0E-4

if __name__ == '__main__':
    unittest.main()
```

## A.3   ITERATIVE CODE CORRECTION

To verify the error correction capability of $k$NN-ECD, we conduct multi-round experiments on $k$NN-ECD, trying to perform iterative code correction on the same datastore. We carry out experiments from the following two aspects. On one hand, we iteratively input the wrong source translations into $k$NN-ECD and compare the overlap of the outputs. The results reveal a significant overlap in corrected translations across multiple trials, with only minor differences in error correction rates ranging from 0.012% to 0.047%. On the other hand, we also attempted to repeatedly feed the wrong output of $k$NN-ECD back into $k$NN-ECD for multiple rounds. The experimental results indicate that only 2.5% of wrong functions are re-corrected in the first round. In subsequent rounds, no more than 0.4% of wrong functions are re-corrected each time. The above experiments illustrate that $k$NN-ECD usually corrects all errors in wrong translation at once, with little additional gain from iterative error correction.

## A.4   STABILITY ANALYSIS.

We analyze the stability of the error correction model in terms of both construction and implementation. In the construction phase, we randomly and evenly divide the error correction training dataset into $m$ sub-datasets, and then generate the corresponding sub_$ECD_{i \in [1,m]}$. In Figure 3, we separately test the independent error correction performance of sub_$ECD_{i \in [1,m]}$ under the $ECS_m$, where $\{$sub_$ECD_1$, sub_$ECD_2$, ..., sub_$ECD_m\}$ show close error correction rates within the same system. Meanwhile, comparing sub_$ECD_{avg}$ of different $k$NN-$ECS_m$ in Figure 5, we find that the larger the sub-datastore, the higher the error correction performance. The above observations indicate that the datastore can stably learn correction information from the error correction dataset during the construction process. Moving on to the implementation phase, we repeatedly feed the same test dataset into $k$NN-ECD, and then compare the overlap of the outputs. The results demonstrate a significant overlap in the corrected translations across multiple trials, with only slight differences in error correction rates ranging from 0.012% to 0.047%. This implies that the error correction model exhibits strong stability during the implementation process, enabling users to trust the output of the model.

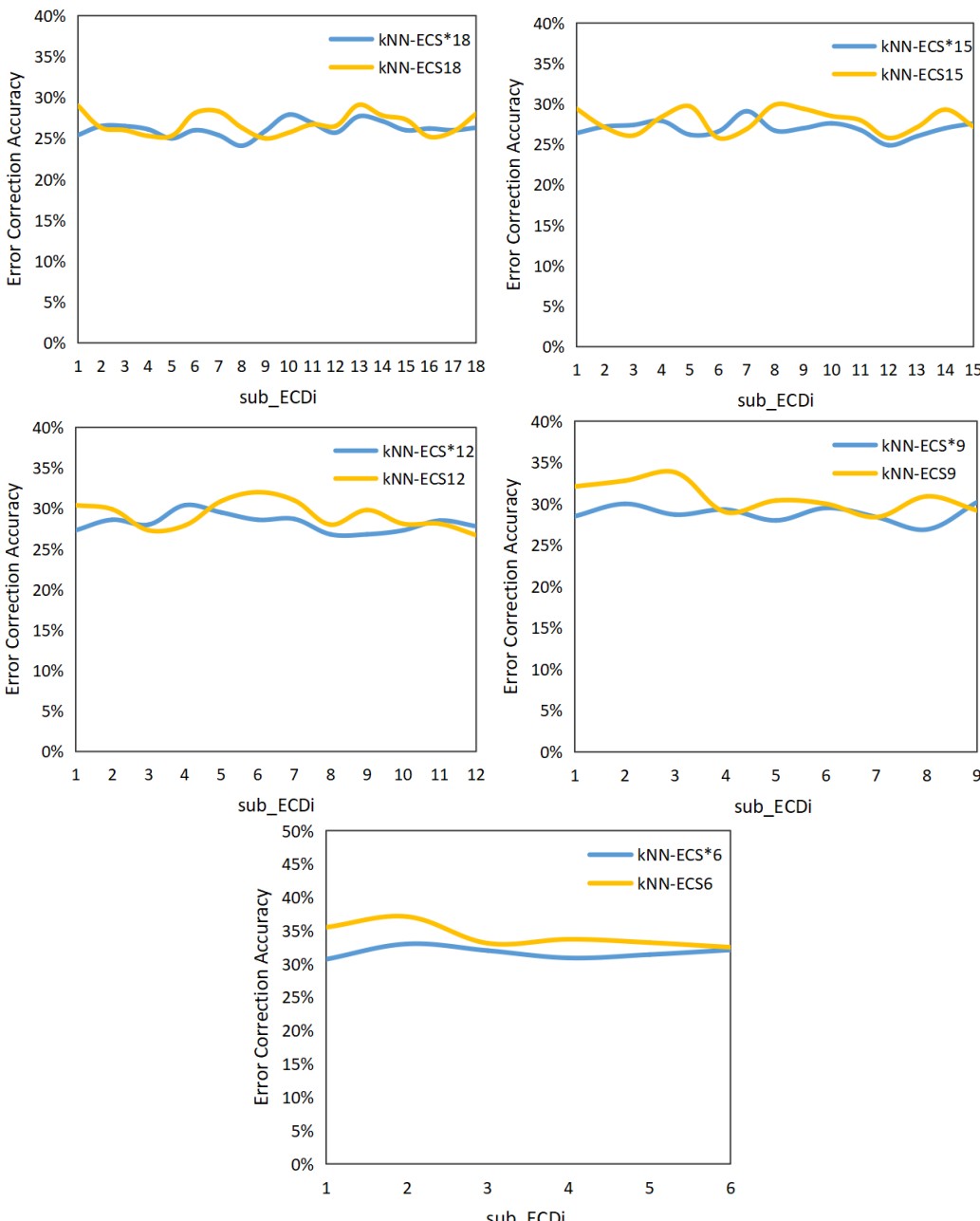

Figure 3: **Independent error correction performance of** $k$**NN-sub_ECD**$_{i \in [1,m]}$ **under** $k$**NN-ECS**$_m$**.** In the same error correction system, where sub_ECDs have similar memory storage, we compare their independent error correction performance respectively. We find that sub_ECDs with approximate information storage capacity exhibit close error correction capabilities. This indicates that datastore can stably learn and apply error correction information from error correction language pairs.

Table 7: The decision-making basis for each generated token in the corrected translation.

| Type | Details |
|---|---|
| Java function | public static void foo ( int [ ] buf )  for ( int i = 0 ; i ¡ buf . length ; i ++ )  buf [ i ] = 7 ; |
| Wrong Python function | def func ( a0 ) : NEW_LINE INDENT for b in a0 : NEW_LINE INDENT b = 7 NEW_LINE DEDENT DEDENT |
| Corrected function | def func ( a0 ) : NEW_LINE INDENT for b in range ( len ( a0 ) ) : NEW_LINE INDENT a0 [ b ] = 7 NEW_LINE DEDENT DEDENT |

| Decision-making basis | |
|---|---|
| | ⟨ 'def func ( a0 , a1 ) : NEW_LINE INDENT for ( b , b ) in enumerate ( a0 ) : NEW_LINE INDENT a0 [ b ] = a1 NEW_LINE DEDENT DEDENT", ''⟩ → 'def' |
| | ⟨ 'def func ( a0 , a1 ) : NEW_LINE INDENT for b in range ( a1 ) : NEW_LINE INDENT yield a0 NEW_LINE DEDENT DEDENT', 'def'⟩ → 'func ' |
| | ⟨ 'def func ( a0 , a1 ) : NEW_LINE INDENT for b in range ( a1 ) : NEW_LINE INDENT yield a0 NEW_LINE DEDENT DEDENT', 'def func'⟩ → '( ' |
| | ⟨ 'def func ( a0 ) : NEW_LINE INDENT b = 0 NEW_LINE for c in a0 : NEW_LINE INDENT b = b + 3 NEW_LINE DEDENT return b NEW_LINE DEDENT', 'def func ('⟩ → 'a0 ' |
| | ⟨ 'def func ( a0 ) : NEW_LINE INDENT sum = 0 NEW_LINE for b in a0 : NEW_LINE INDENT sum += b NEW_LINE DEDENT c = d = sum NEW_LINE for b in a0 : NEW_LINE INDENT d -= b NEW_LINE if c == d : NEW_LINE INDENT return b NEW_LINE DEDENT c += b NEW_LINE DEDENT return - 1 NEW_LINE DEDENT', 'def func ( a0'⟩ → ') ' |
| | ⟨ 'def func ( a0 ) : NEW_LINE INDENT b = 0 NEW_LINE for c in a0 : NEW_LINE INDENT b += c NEW_LINE DEDENT return sum NEW_LINE DEDENT', 'def func ( a0 )'⟩ → ': ' |
| | ⟨ 'def func ( a0 ) : NEW_LINE INDENT for ( b , c ) in enumerate ( a0 ) : NEW_LINE INDENT if not c . isalnum ( ) : NEW_LINE INDENT return b - 1 NEW_LINE DEDENT DEDENT return len ( a0 ) - 1 NEW_LINE DEDENT', 'def func ( a0 ) :'⟩ → 'NEW_LINE ' |
| | ⟨ 'def func ( a0 , a1 ) : NEW_LINE INDENT for b in a0 : NEW_LINE INDENT b += a1 NEW_LINE DEDENT DEDENT', 'def func ( a0 , a1 ) : NEW_LINE'⟩ → 'INDENT ' |
| | ⟨ 'def func ( a0 , a1 ) : NEW_LINE INDENT for b in a0 : NEW_LINE INDENT b += a1 NEW_LINE DEDENT DEDENT', 'def func ( a0 , a1 ) : NEW_LINE INDENT'⟩ → 'for ' |
| | ⟨ 'def func ( a0 , a1 ) : NEW_LINE INDENT for b in a0 : NEW_LINE INDENT b += a1 NEW_LINE DEDENT DEDENT', 'def func ( a0 , a1 ) : NEW_LINE INDENT for'⟩ → 'b ' |
| | ⟨ 'def func ( a0 , a1 ) : NEW_LINE INDENT for b in a0 : NEW_LINE INDENT b += a1 NEW_LINE DEDENT DEDENT', 'def func ( a0 , a1 ) : NEW_LINE INDENT for b'⟩ → 'in ' |
| | ⟨ 'def func ( a0 , a1 ) : NEW_LINE INDENT for b in a0 : NEW_LINE INDENT b += a1 NEW_LINE DEDENT DEDENT', 'def func ( a0 , a1 ) : NEW_LINE INDENT for b in'⟩ → 'range ' |
| | ⟨ 'def func ( a0 , a1 ) : NEW_LINE INDENT for b in a0 : NEW_LINE INDENT b += a1 NEW_LINE DEDENT DEDENT', 'def func ( a0 , a1 ) : NEW_LINE INDENT for b in range'⟩ → '( ' |
| | ⟨ 'def func ( a0 ) : NEW_LINE INDENT sum = 0 NEW_LINE for b in a0 : NEW_LINE INDENT sum += b NEW_LINE DEDENT c = d = sum NEW_LINE for b in a0 : NEW_LINE INDENT d -= b NEW_LINE if c == d : NEW_LINE INDENT return b NEW_LINE DEDENT c += b NEW_LINE DEDENT return - 1 NEW_LINE DEDENT', 'def func ( a0 ) : NEW_LINE INDENT sum = 0 NEW_LINE for b in range ('⟩ → 'len ' |
| | ⟨ 'def func ( a0 ) : NEW_LINE INDENT sum = 0 NEW_LINE for b in a0 : NEW_LINE INDENT sum += b NEW_LINE DEDENT c = d = sum NEW_LINE for b in a0 : NEW_LINE INDENT d -= b NEW_LINE if c == d : NEW_LINE INDENT return b NEW_LINE DEDENT c += b NEW_LINE DEDENT return - 1 NEW_LINE DEDENT', 'def func ( a0 ) : NEW_LINE INDENT sum = 0 NEW_LINE for b in range ( len'⟩ → '( ' |
| | ⟨ 'def func ( a0 , a1 ) : NEW_LINE INDENT for b in a0 : NEW_LINE INDENT b += a1 NEW_LINE DEDENT DEDENT', 'def func ( a0 , a1 ) : NEW_LINE INDENT for b in range ( len ('⟩ → 'a0 ' |
| | ⟨ 'def func ( a0 ) : NEW_LINE INDENT sum = 0 NEW_LINE for b in a0 : NEW_LINE INDENT sum += b NEW_LINE DEDENT c = d = sum NEW_LINE for b in a0 : NEW_LINE INDENT d -= b NEW_LINE if c == d : NEW_LINE INDENT return b NEW_LINE DEDENT c += b NEW_LINE DEDENT return - 1 NEW_LINE DEDENT', 'def func ( a0 ) : NEW_LINE INDENT sum = 0 NEW_LINE for b in range ( len ( a0'⟩ → ') ' |
| | ⟨ 'def func ( a0 ) : NEW_LINE INDENT sum = 0 NEW_LINE for b in a0 : NEW_LINE INDENT sum += b NEW_LINE DEDENT c = d = sum NEW_LINE for b in a0 : NEW_LINE INDENT d -= b NEW_LINE if c == d : NEW_LINE INDENT return b NEW_LINE DEDENT c += b NEW_LINE DEDENT return - 1 NEW_LINE DEDENT', 'def func ( a0 ) : NEW_LINE INDENT sum = 0 NEW_LINE for b in range ( len ( a0 )'⟩ → ') ' |
| | ⟨ 'def func ( a0 , a1 ) : NEW_LINE INDENT for b in a0 : NEW_LINE INDENT b += a1 NEW_LINE DEDENT DEDENT', 'def func ( a0 , a1 ) : NEW_LINE INDENT for b in range ( len ( a0 ) )'⟩ → ': ' |
| | ⟨ 'def func ( a0 , a1 ) : NEW_LINE INDENT for b in a0 : NEW_LINE INDENT b += a1 NEW_LINE DEDENT DEDENT', 'def func ( a0 , a1 ) : NEW_LINE INDENT for b in range ( len ( a0 ) ) :'⟩ → 'NEW_LINE ' |
| | ⟨ 'def func ( a0 , a1 ) : NEW_LINE INDENT for b in a0 : NEW_LINE INDENT b += a1 NEW_LINE DEDENT DEDENT', 'def func ( a0 , a1 ) : NEW_LINE INDENT for b in range ( len ( a0 ) ) : NEW_LINE'⟩ → 'INDENT ' |
| | ⟨ 'def func ( a0 , a1 ) : NEW_LINE INDENT for b in a0 : NEW_LINE INDENT b += a1 NEW_LINE DEDENT DEDENT', 'def func ( a0 , a1 ) : NEW_LINE INDENT for b in range ( len ( a0 ) ) : NEW_LINE INDENT'⟩ → 'a0 ' |
| | ⟨ 'def func ( a0 , a1 ) : NEW_LINE INDENT for b in a0 : NEW_LINE INDENT b += a1 NEW_LINE DEDENT DEDENT', 'def func ( a0 , a1 ) : NEW_LINE INDENT for b in range ( len ( a0 ) ) : NEW_LINE INDENT a0'⟩ → '[ ' |
| | ⟨ 'def func ( a0 , a1 ) : NEW_LINE INDENT for b in a0 : NEW_LINE INDENT b += a1 NEW_LINE DEDENT DEDENT', 'def func ( a0 , a1 ) : NEW_LINE INDENT for b in range ( len ( a0 ) ) : NEW_LINE INDENT a0 ['⟩ → 'b ' |
| | ⟨ 'def func ( a0 , a1 ) : NEW_LINE INDENT for b in a0 : NEW_LINE INDENT b += a1 NEW_LINE DEDENT DEDENT', 'def func ( a0 , a1 ) : NEW_LINE INDENT for b in range ( len ( a0 ) ) : NEW_LINE INDENT a0 [ b'⟩ → '] ' |
| | ⟨ 'def func ( a0 ) : NEW_LINE INDENT [ a0 ] = 0 NEW_LINE DEDENT', 'def func ( a0 ) : NEW_LINE INDENT a0 [ 0 ]'⟩ → '= ' |
| | ⟨ 'def func ( a0 ) : NEW_LINE INDENT b = a0 . find ( b' ¿ ' ) NEW_LINE c = 7 NEW_LINE return a0 [ c : b ] NEW_LINE DEDENT ¡/s¿", "def func ( a0 ) : NEW_LINE INDENT b = a0 . find ( ' ¿ ' ) NEW_LINE c = ' ⟩ → '7 ' |
| | ⟨ 'def func ( a0 = None ) : NEW_LINE INDENT return 8 if a0 is None else 7 NEW_LINE DEDENT', 'def func ( a0 ) : NEW_LINE INDENT return 8 if a0 == 256 else 7'⟩ → 'NEW_LINE ' |
| | ⟨ 'def func ( a0 , a1 ) : NEW_LINE INDENT for b in a0 : NEW_LINE INDENT b += a1 NEW_LINE DEDENT DEDENT', 'def func ( a0 , a1 ) : NEW_LINE INDENT for b in range ( len ( a0 ) ) : NEW_LINE INDENT a0 [ b ] += a1 NEW_LINE'⟩ → 'DEDENT ' |
| | ⟨ 'def func ( a0 , a1 ) : NEW_LINE INDENT for b in a0 : NEW_LINE INDENT b += a1 NEW_LINE DEDENT DEDENT', 'def func ( a0 , a1 ) : NEW_LINE INDENT for b in range ( len ( a0 ) ) : NEW_LINE INDENT a0 [ b ] += a1 NEW_LINE DEDENT'⟩ → 'DEDENT ' |
| | ⟨ 'def func ( a0 , a1 ) : NEW_LINE INDENT for b in a0 : NEW_LINE INDENT b += a1 NEW_LINE DEDENT DEDENT', 'def func ( a0 , a1 ) : NEW_LINE INDENT for b in range ( len ( a0 ) ) : NEW_LINE INDENT a0 [ b ] += a1 NEW_LINE DEDENT DEDENT'⟩ → 'EOS ' |

* ⟨·, ·⟩ → '* ' : ⟨·, ·⟩ represents the decision-making basis for each generated token '* ' in the corrected Python function.

* NEW_LINE, INDENT, DEDENT represent newline, indentation, dedent in code formatting, respectively.

* EOS: end of sentence.

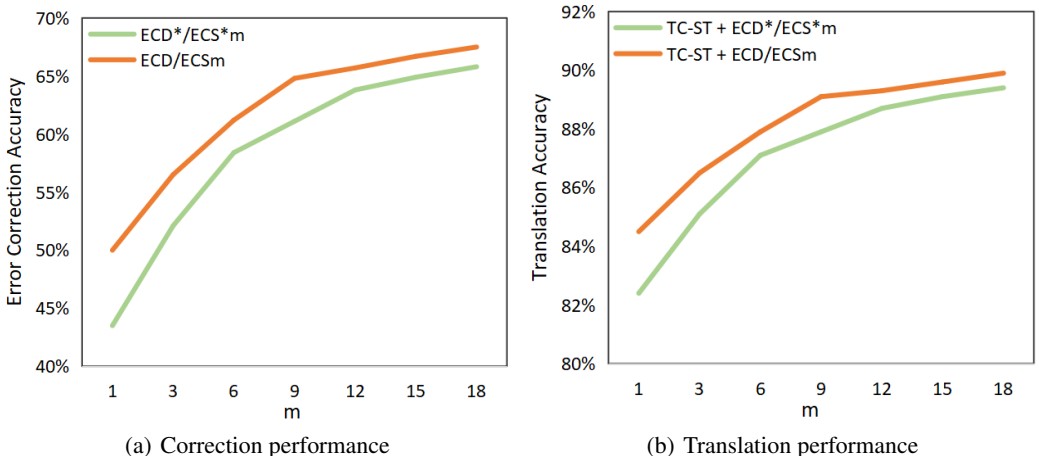

(a) Correction performance          (b) Translation performance

Figure 4: **The influence of diverse datastore distributed structures.** We find that the distributed architecture can effectively enhance the error correction performance by efficiently learning and applying massive error correction information, thereby improving translation accuracy. Meanwhile, we also explore the impact of the unified name rule on the error correction model. This approach exhibits a positive gain effect, effectively improving the model's error correction capabilities.

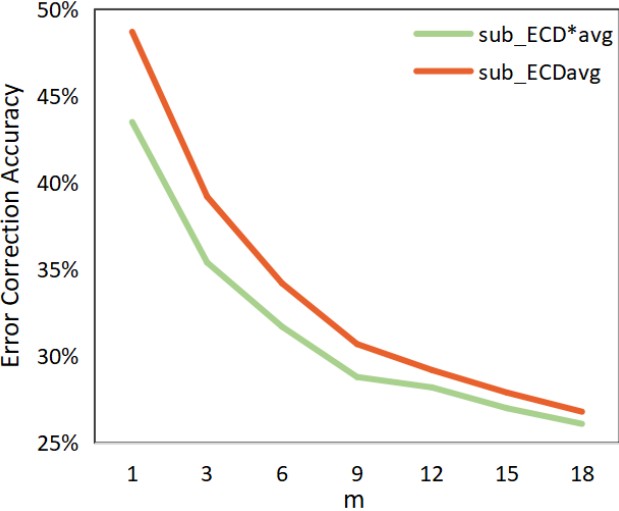

Figure 5: **The influence of datastore capacity on translation accuracy.** We observe a positive correlation between the sub-datastore capacity and error correction performance. The larger the sub-datastore, the higher the error correction capability of the sub-datastore. This finding suggests that the datastore can robustly acquire and integrate correction information from the error correction language pairs.

