# OpenReview forum: "An interpretable error correction method for enhancing code-to-code translation"
_ICLR.cc/2024/Conference — ICLR 2024 poster_

### Official Review · Reviewer_7Q1Z · 2023-10-15

**Soundness:** 3 good
**Presentation:** 3 good
**Contribution:** 2 fair
**Rating:** 6
**Confidence:** 4

**Summary:**

This paper focuses on improving Java $\rightarrow$ Python translation using error correction, rather than retraining the underlying translation model. They devise two error correction techniques (kNN-ECD and kNN-ECS) based on kNN-MT, which entails retrieving from datastores. To build this datastore, they first collect 82,665 Java functions and generate high-quality unit tests for them using EvoSuite. Then, they use TransCoder-ST to translate the Java functions paired with the unit tests to Python. From these they extract pairs of the form (failed Python function, successful Python function), which are then used to build (a) datastore(s). The datastore is organized based on two components: (1) (key, value) pairs and (2) (key, value) $\rightarrow$ token. The query to this datastore is formed by using the last decoder hidden states corresponding to the full source input (i.e., failed Python function) and partial target (i.e., possible correction generated so far). To reduce noise caused by diverse rare identifiers during retrieval, they apply the unified name rule. In kNN-ECD, only one round of correction is performed. In kNN-ECS_{m}, they perform $m$ rounds of correction, with m smaller datasets (after segmenting the large datastore into $m$ parts). Results show that kNN-ECS outperforms kNN-ECD as well as a vanilla TransCoder-ST with no error correction.

**Strengths:**

- The proposed approach successfully corrects errors to a certain extent, without retraining the model or re-sampling the model many times, which is usually done in self-repair.
- The idea of multi-round error correction and the analysis done with this, varying the number of rounds, and analyzing the performance for each of these, is quite interesting and may inspire future work.

**Weaknesses:**

- Evaluation is based on translated unit tests generated by the same model that the authors are trying to correct translation errors for. Therefore, the unit tests that are generated could be wrong, and so the evaluation is unreliable. Evaluation should be performed based on a separate, high-quality set of unit tests. Possibly datasets like HumanEval-X would be better alternatives here.
- The experiments and results are fairly limited. First, the authors focus on only Java $\rightarrow$ Python and fail to consider other languages or even the reverse direction of Python $\rightarrow$ Java. Next, Table 1 seems to be missing many baselines and other models to which their approach should be compared. Namely, the only baseline is the pure TransCoder-ST model, which is only the starting point of their approach. The authors discuss that the main advantage of their approach is that no retraining is required, so it would be important to see how their approach performs relative to a retraining-based one. For this, they could have simply fine-tuned TransCoder-ST on the error correction pairs they collected for building their datastore. Next, latency is not measured, even though the authors discuss latency in related work. It seems that retrieving from a large datastore or retrieving multiple times from smaller datastores could take a long time, so it would be important to understand how the overall latency compares to other approaches. Finally, the authors do not report results on state-of-the-art code models, so it is difficult to assess the true value of their approach.
- The authors present the unified name rule as a novelty; however, I do not find this to be that novel, given the work the authors discussed in the "Related Work" section.
- There are multiple aspects of the paper that are not clear.  Please see the "Questions" section.

**Questions:**

1) Based on what is written in the paper, 10 Python functions with unit test cases are generated for each Java function. So, you have $(func_1, tests_1), (func_2, tests_2), (func_3, tests_3)... (func_{10}, tests_{10})$. Success is measured by executing the tests in some Python environment, where $func_i$ is considered a success if it passes all tests in $tests_i$. By this definition, suppose $func_1$ fails $tests_1$ and $func_2$ passes $tests_2$.  The paper states "we combined the first failed Python function with the first successful Python function to form an error correction language pair." Based on this, it seems that $(func_1, func_2)$ would be considered an error correction pair. However, there is no guarantee that $tests_1 = tests_2$, meaning that the two functions could be executed against different test suites. Therefore, $func_2$ may not actually correspond to a correction of $func_1$. Could this please be clarified?
2) The "interpretable decision-making" idea is not clear to me. It seems that you are suggesting that the reasoning for predicting a specific token at a timestep $t$ can be attributed to the source and partial target function predicted so far. This is also the case for transformer-based decoders, so it is not clear to me how your approach can be considered more interpretable than a transformer as they claim.
3) In 3.2, you state that the hidden representations from the last layer of the decoder are used to build the (key,value) and query. My understanding is that the (key ,value) and query correspond to (failed Python function, partial Python function). It is not clear to me how there would be a decoder state corresponding to the failed Python function since that is passed into the encoder (Figure 1). Or is (failed Python function, partial Python function) meant to actually only represent the representation of the partial Python function generated so far, as labeled as "Key" in Figure 1?
4) You claim that the improvement of ECS over ECD is "primarily attributed to its distributed structure, which includes diverse datastore variants." However, you do not seem to have multi-round experiments with ECD in which you repeatedly perform retrieval/correction on the same large datastore up to $m$ times. Therefore, isn't it possible that the advantage is actually from doing iterative code correction rather than the distributed nature of it?

---

> ### Author Response · Authors · 2023-11-20
> **Response to Reviewer 7Q1Z**
>
> Thank you very much for your valuable comments and effort in improving this submission.  We will release the code, models, test dataset, training dataset and unit test dataset in the future.
>
> Response to Weaknesses_1:
>
> For this question, we add detailed explanations and experiments in the revised paper (Please refer to APPENDIX A.2 'GENERATION OF PYTHON UNIT TEST DATASET' and Table 6).
>
> Response to Weaknesses_2:
>
> (1) The current paper is a completely new approach in the field of program translation. Although the scope of our research is limited, we conducted detailed experiments to demonstrate the feasibility of this research. We appreciate your question about expanding the model to a multilingual environment, and this is precisely what we are currently working on.
>
> (2) we provide other more comprehensive baseline comparisons in the revised paper (Please refer to Table 1).
>
> (3) Thank you very much for your suggestion, we will try to fine-tune TransCoder-ST using error correction datasets in future work.
>
> (4) For the improvement and traditional approaches on $k$NN-MT, the generated results require both retrieving the datastore and running the Transformer model. However, in our work, the results of the error correction model are generated only by retrieving $(\text{key}, \text{value})$ pairs from the datastore, without incorporating the Transformer model.
>
> For $k\text{NN-ECD}$ (with the shortest processing time), we processed 2017 test samples with a total runtime of 520.28 seconds (average processing time of 0.26 seconds per sample). The translation accuracy achieved under this method was 84.5\%.
>
> For $k\text{NN-ECS}_{18}$ (with the longest processing time), we executed 2017 test samples with a total runtime of 10207.46 seconds (average processing time of 5.06 seconds per sample). The corresponding translation accuracy for this method was 89.9\%.
>
> Response to Weaknesses_3:
>
> (1) In previous works, researchers have ignored the possibility of unifying diverse identifiers in both training dataset and test dataset. In addition, existing identifier replacement methods are extremely simple, often breaking the code structure and neglecting to avoid replacing special identifiers such as built-in functions. Furthermore, there are no specific experimental results to illustrate the impact of unifying identifiers.
>
> (2) We systematically propose the work of unifying all identifiers of training and test datasets in the field of program translation, while providing detailed experimental results. Our proposed method performs targeted replacements for diverse function names, variable names, and parameter names without disrupting the code structure. During this process, we do not replace built-in name/ module_name/ given parameter.
>
> Response to Questions_1:
>
> For multiple Python functions under the same Java function, we use the same unit test dataset, and almost unit test dataset typically contains 3$\sim$6 unit tests. That is to say, $func_1$, $func_2$,..., $func_{10}$ use the same unit test dataset, where this unit test dataset usually contains 3$\sim$6 unit tests. (If a Python function can pass all tests under the unit test dataset, it succeeds; otherwise, it fails.)
>
> Response to Questions_2:
>
> In the newly submitted paper, we add detailed experiment and explanation in Section 4.2 'Interpretability of error correction model' and Table 7.
>
>
> Response to Questions_3:
>
> (1) Is (failed Python function, partial successful Python function) meant to actually only represent the representation of the partial Python function generated so far, as labeled as "Key"? Yes. The entire process is similar to the Transformer training process. The key is the output of the cross-attention layer in the decoder part. Reference [1] explains the technical details.
>
> (2) More specifically, the last hidden layer (i.e. cross-attention layer) is the coding (i.e. key) of <src\_func, tgt\_func\_pref >, where src\_func represents the failed Python function, and tgt\_func\_pref comes from the successful Python function. In the process of generating the key, the wrong Python function of the error correction language pair is the input of the encoder side, and the correct Python function of the error correction language pair is the input of the decoder side.
>
> Response to Questions_4:
>
> We provide detailed descriptions and experimental data in the revised paper (Please refer to  Appendix A.3 'ITERATIVE CODE CORRECTION').
>
> [1]Khandelwal U, Fan A, Jurafsky D, et al. Nearest neighbor machine translation[J]. arXiv preprint arXiv:2010.00710, 2020.

---

> > ### Comment · Reviewer_7Q1Z · 2023-11-21
> >
> > Thank you for providing detailed answers to my questions and for making changes to the manuscript. I have read the other reviews and I have slightly increased my score. While the quality of the paper has definitely improved with the additional experiments and analyses, I think this work could still use another revision with some more of the experiments mentioned in the reviews.

---

> > > ### Author Response · Authors · 2023-11-21
> > > **Response to Reviewer 7Q1Z**
> > >
> > > Thank you so much for your encouragement! We will add relevant experiments to the subsequent revised paper. Thank you again for your suggestions on our paper.

---

### Official Review · Reviewer_oueh · 2023-11-01

**Soundness:** 3 good
**Presentation:** 3 good
**Contribution:** 3 good
**Rating:** 6
**Confidence:** 3

**Summary:**

To address a need for code-to-code translation accuracy improvements, the authors propose to combine a transformer model, specifically TransCoder-ST, with an error correction model that optionally overwrites the output of the transformer model. They do so in two ways. Initially, they consider the error correction model to be a single error-correction datastore in which they perform kNN (kNN-ECD). Later, they improve on the initial model by dividing the dataset into m-sub-datasets and they construct a distributed datastore (kNN-ECD$_m$). To make the dataset more uniform, the authors also employ a "unified name rule", to perform $\alpha$-renaming while keeping certain type information. They show that this full-pipeline can improve TransCoder-ST performance on Java to Python translation from 68.9% to 89.9%.

**Strengths:**

- Simple framework both in the single and multi-round error correction.
- Shows generalisation of fixing patterns (and the authors check for data leakage).
- Extensive ablation to understand which pipeline components contribute to the overall performance (single- vs multi-round error correction, sub-datastore performance, unified renaming)

**Weaknesses:**

- Interpretability feels like an after-thought, it is left to the reader to infer it from the use of kNN-MT derived methods. Indeed, the discussion in S4.2 focuses more on the ability of the model to generalise from the examples (which is interesting and significant), but the showing the mapping to a software engineer would do little to explain the model decision and gain trust in the model. To build on the S4.2 example, a user explanation would be that the final values should be "int", and it is difficult to derive this insight from the mapping.

- On dataset construction, EvoSuite will overfit to the current implementation, which means there is an underlying assumption that the Java version is bug free. Further, translation of the PyTests from Java Unit tests can also be erroneous.

**Questions:**

In the multi-round error correction(kNN-ECD$_m$), does each subsequent ECD get the mis-corrected version from the previous module or does each ECD get the original wrong translation?
If the former, does kNN-ECD$_m$ perform partial corrections that build on top of each other?

On the dataset construction, have PyTest translations been manually or otherwise been validated to be correct/faithful to the EvoSuite output?

---

> ### Author Response · Authors · 2023-11-20
> **Response to Reviewer oueh**
>
> Thank you very much for your valuable and insightful comments.
>
> Response to Weaknesses_1:
>
> (1) We apologize for any confusion caused. Due to space limitations and the desire to present important experimental results in the main text, we merged the interpretability experiment and the generalization experiment. In the revised paper, we restate the two experiments separately and provide more detailed experimental data (Please refer to Section 4.2 'Generalizability analysis' and Section 4.2 'Interpretability of error correction model').
>
> (2) Indeed, interpretability was not an afterthought. It was an interesting discovery we made during the initial exploration of the $k$NN-MT method in the field of program translation. Since the $(\text{key}, \text{value})$ pair is a string of numbers, it is difficult for us to obtain intuitive information from it. In such cases, the uncoded form of the key-value pair $ \langle  \cdot, \cdot \rangle$ $\to$  '//' can provide more clear information, where the coding of $ \langle  \cdot, \cdot \rangle$ is key, and the coding of '//' is value.  In the newly submitted paper, we add detailed experiments and explanations in Section 4.2 'Interpretability of error correction model' and Table 7.
>
> Response to Weaknesses_2:
>
> For this issue, we provide a detailed explanation and experimental data in the revised paper  (Please refer to APPENDIX A.2 `GENERATION OF PYTHON UNIT TEST DATASET’ and Table 6).
>
> Response to Questions_1:
>
> In $k\text{NN-ECS}_{m}$, the input of each sub\_ECD is the original wrong translation. At the beginning of the experiment, we also considered the method you mentioned, using the wrong output from the previous sub\_ECD as the input for the next sub\_ECD. However, the experimental performance was very unsatisfactory.
>
> Response to Questions_2:
>
> Yes, this is one of the important prerequisites to ensure the accuracy of the experimental results.  For Python error correction language pairs, each wrong function in the training dataset and test dataset has a corresponding correct function (ground truth) that passes all unit tests. This guarantees that if the wrong function can be corrected, it can also pass all unit tests. Furthermore, for each Python function, its unit test dataset usually contains 3$\sim$6 unit tests.
> For this question, we add a detailed explanation and experiment in the revision of the paper (Please refer to APPENDIX A.2 'GENERATION OF PYTHON UNIT TEST DATASET' and Table 6).

---

> ### Comment · Reviewer_oueh · 2023-11-21
>
> Thank you for the responses and clarifications!
>
> I feel that the restructured section 4.2 now more clearly delineates between the interpretability and generalisation argument. I also think there was a misunderstanding on my end where I considered interpretability as an SE practitioner rather than an ML researcher/engineer. I think Table 7 now better demonstrates the author's argument for interpratability.
>
> Regarding Q1, thank you for the clarification.
>
> Regarding Q2, the link to the source code answers my question. The fact that EvoSuite produces a very specific kind of unit test structure is exploited to pattern-match translate the unit tests to functionally equivalent Python tests. There are other concerns I have with this approach, specifically under-sampling of the program behaviour space, but these are fundamental to the approach and cannot be corrected/should be accepted as limitations or a trade-off.
>
> I maintain my score, but I think the paper is improved in clarity with the additional information in the appendix.

---

> > ### Author Response · Authors · 2023-11-21
> > **Response to Reviewer oueh**
> >
> > Thank you very much for your reply. Regarding your concern about Q2, we have considered it in the early stage of our experiment.
> >
> > (1) Here are some of our experimental data: the amount of Java functions in the sampling space is 12,312,432; the amount of filtered Java functions is 82,665; the amount of error correction language pairs is 21,385; and the amount of  $(\text{key}, \text{value})$ pairs in ECD is 937,362. (We will release the code, models, test dataset, training dataset and unit test dataset in the future.) dataset, training dataset and unit test dataset in the future.)
> >
> > (2) We had used a larger error correction dataset to generate the datastore (containing 88,056 error correcting language pairs). However, the improvement in error correction performance is very small. That is, the current data sample size is sufficient for correcting common errors.
> >
> > (3)Finally, the datastore is scalable, and new error correction knowledge can be learned by adding new error correction information directly to the datastore, and the implementation method is extremely simple and efficient.  In other words, for a wrong output, we can correct this type of error by adding repair information (i.e.  $(\text{key}, \text{value})$ pair), directly to the datastore. This aspect is part of our future work and has been validated.

---

> > > ### Comment · Reviewer_oueh · 2023-11-21
> > >
> > > Thank you for this clarification, but you misunderstood what sample space I meant. That is fine as I now realise my usage is more common outside ML.
> > >
> > > Unit tests exercise a very limited set of behaviours for programs under test. In a sense, unit tests sample potential inputs and verify the output (and potential intermediate points) align with expected values. To ensure behavioural equivalence would require a proof (in the sense of Coq or Agda for example). It is this program behaviour space that I meant when saying that the space is under-sampled, which is inherent to the use of unit tests. (Note, I am aware that you use a separate definition of functional equivalence which is parametrised by the test suite).

---

> > > > ### Author Response · Authors · 2023-11-21
> > > > **Response to Reviewer oueh**
> > > >
> > > > Thanks for the clarification. Indeed, this is a point that is often overlooked in the field of program translation. We will take this into consideration in our subsequent experiments. Once again, we appreciate your valuable suggestion!

---

### Official Review · Reviewer_Uzpj · 2023-11-01

**Soundness:** 2 fair
**Presentation:** 2 fair
**Contribution:** 3 good
**Rating:** 6
**Confidence:** 3

**Summary:**

The paper proposes an error correction method, KNN-ECD, which is based on KNN-MT and improves the performance of code translation.  Building upon this, the paper further propose $kNN-ECS_{m}$, which divides the data store to $m$ sub-datastores. In addition, the paper proposes a new unified name rule to encourage the datastore to focus more on code logic and structure rather than diverse rare identifiers. The experiments show the the proposed methods largely improve the translation accuracy.

**Strengths:**

1. The paper applies the $kNN-MT$ to the code translation and obtain a significant improvement of the translation accuracy. Using functional equivalence as evaluation metrics instead of BLEU better reflects the true  code translation quality. And the proposed method increase the accuracy by about 20%.

2. The paper proposes a novel $kNN-ECS_{m}$ framework, which further improves the translation accuracy of program.

3. The paper performance extensive empirical analysis of the proposed method.

**Weaknesses:**

1. $kNN-ECD$ is very similar to $kNN-MT$. Therefore, the technical contribution of the paper is limited.

2. The motivation of applying $kNN-MT$ is not very clear. Although $kNN-MT$  is useful for natural language translation, is there some particular reasons that it will be more effective for programming languages.

3. The presentation is experiment results is hard to read, especially for Table 3 and Table 4. I would suggest the authors to use Figures to present this results and put the detailed numbers in the Appendix.

4. The paper does not show the proposed method can perform error correction for OOD errors. The paper uses model $A$ to build the pair of incorrect programs and correct programs. Therefore, the error is specifically related to model $A$ itself. For a new model $B$, it may make different kinds of error, does the proposed method with learning datastore for model $A$ can fix the error of model $B$. If not, the method requires building datastore for every new method, which largely limiting the application of the proposed method.


Minor:

"Uncorrect" should be changed into "Incorrect"

**Questions:**

1. For unified name rule, how to identify the variable name or function name in a program and replace them? Is it replaced by some automatic tools?

2. How do the method judge the wrong translations of TransCoder-ST? Is the error correction only applied the wrong programs?

3. What does the method have better interpretability? The key value pairs in the datastore is still based on neural networks.

4. Is there any fine-tuning stage of the TransCoder model?

---

> ### Author Response · Authors · 2023-11-20
> **Response to Reviewer Uzpj**
>
> Thank you for providing high-quality feedback on our work. In addition, thank you for pointing out some presentation problems in the paper. We will correct these issues in a future revision of the paper.
>
> Response to Weaknesses_1&2:
>
> (1) As you pointed out, our research goal is to significantly improve translation performance without retraining the large Transformer model. In our preliminary work, we found that the use of $k$NN-MT technology indeed provides substantial technical support. During this process, we were surprised to find that the $k$NN-MT method can provide interpretable results, but this point has not been thoroughly explored in previous research.
>
> (2) Regarding the interpretability, we add a detailed explanation and experiment in the revised paper (Please refer to  Section 4.2 `Interpretability of error correction model’ and Table 7).
>
> Response to Weaknesses_3:
>
> Thank you very much for your suggestions.  In the revision of the paper, we restate the experimental results more clearly and address the grammatical and expression problems. (please refer to  APPENDIX Figures 3-5)
>
>
> Response to Weaknesses_4:
>
> (1) Your suggestion is also one of the key issues we focused on in the early stages of our work. Firstly, our model is generalizable (please refer to Section 4.2 `Generalizability analysis’).
>
> (2) Secondly, since we were curious about which  errors could be corrected, we roughly classified translation error types in the initial experiments and found that errors could be categorized into fixed major types. In addition, kNN-ECD and kNN-ECSm contain 937362 $(\text{key}, \text{value})$ pairs, which include enough error correction information.
>
> (3) Finally, the datastore is scalable, and new error correction knowledge can be learned by adding new error correction information directly to the datastore, and the implementation method is extremely simple and efficient. This aspect is part of our future work and has been validated.
>
> Response to Questions_1:
>
> Yes. First, we parse the code into an Abstract Syntax Tree (AST) using the Python standard library module ‘ast’. We then traverse the AST and replace corresponding function names, variable names, and parameter names with new names. In this process, we define is_builtin_name/ is_module_name/ is_parameter_name in the code to check if the name is builtin_name/ module_name/ given_parameter. If it is, we do not replace it.
>
> Response to Questions_2:
>
> Indeed, we use unit testing to check whether the translation of TransCoder-ST [1] is correct. This is what the author of TransCoder-ST uses to verify whether the output is correct. Then, we correct the wrong translation of TransCoder-ST.
>
> Response to Questions_3:
>
> (1) In the newly submitted paper, we add explanation and experimental details in Section 4.2 `Interpretability of error correction model’ and Table 7.
>
> (2) The generation of $(\text{key}, \text{value})$ pairs is indeed based on the Transformer model. For each token generated by the Transformer-based model, the traditional Transformer model cannot track and identify which snippets in the training dataset contributed to the output token. However, $k\text{NN-ECD}/k\text{NN-ECS}_{m}$ can address this issue, providing an intuitive and readable decision-making basis for each output token. We consider this an advantage compared to the Transformer model.
>
> Response to Questions_4:
>
> There is no fine-tuning process during the construction of the datastore.
>
> [1]Roziere B, Zhang J M, Charton F, et al. Leveraging automated unit tests for unsupervised code translation[J]. arXiv preprint arXiv:2110.06773, 2021.

---

> ### Author Response · Authors · 2023-11-20
> **Revised Manuscript Submission**
>
> Dear Reviewers,
>
> We greatly appreciate the valuable suggestions provided by the reviewers for our paper. We have added experiments and analyses to the revised manuscript, which has been resubmitted. Due to time constraints, some experiments will be supplemented in subsequent work.
>
> Kind regards,
>
> the Authors

---

> ### Comment · Reviewer_Uzpj · 2023-11-23
> **Post-rebuttal comments**
>
> I appreciate the efforts of the authors in the rebuttal. I helps to resolve most concerns. However, I still have the following concerns, and I lean to keep my score unchanged.
>
> - Weakness 1 & 2:
>
>   What I am arguing is that the technical method of this paper is very similar to previous paper of KNN-MT. And I do not find the authors' response help to resolve the concern. Besides, the authors' response cannot provide more insight about why KNN-MT is particularly useful in program translation.
>
> - Weakness 4
>
>   The revised version does not provide numerical evidence about the generalizability. I think the best way is to use a different model to build a dataset with different kinds of error and evaluate the proposed method in this new dataset.
>
> - Questions_2
>
>   If so, the unit tests are required for the error correction, which significantly limit the application of the proposed method. I suggest the author to show that whether applying the proposed method to all programs (not limited to wrongly translated programs) still results in large improvements.

---

> ### Author Response · Authors · 2023-11-23
> **Response to Reviewer Uzpj**
>
> Response to Weaknesses_1&2:
>
> Thanks for your response. The work of $k\text{NN-ECD}$
>  is indeed based on $k$NN-MT, but the two are slightly different.
>
> (1) The results generated by $k$NN-MT need to be combined with datastore and Transformer at the same time. In this case, it is impossible to return an explainable decision-making basis for each output token.
>
> (2) Secondly, the results generated by $k\text{NN-ECD}$
>  only use datastore, which can directly return an interpretable decision-making basis for each output token. (see Table 7)
>
> (3) Regarding the use of decision-making basis: We can see the decision-making basis used by the uncorrected error type. This type of error can be corrected by replacing or deleting related  $(\text{key}, \text{value})$ pairs in datastore, where decision-making basis is the coding of key.
>
>
> Response to Weaknesses_4:
>
> (1) We understand your concerns very much. The results in Table 4 can provide some explanation for the generalization analysis. Under $k\text{NN-ECS}_{m}$,  each sub_ECD is generated from a completely different error correction dataset (with the same number of error correction pairs), but they all show similar performance on the same testset.The decision-making results (in Table 2) can also also provide the explanation for generalization.
>
> (2) We will release the code, models, test dataset, training dataset and unit test dataset in the future. This is also the data support for the dataset comparison experiment mentioned in our generalization analysis.
>
>  If the above experiments cannot solve your problem, we will supplement relevant experiments in the future.
>
>
> Response to Questions_2:
>
> (1) In real life, users often put the output of the translation model directly into their compiler for testing. If it is wrong, the user inputs the wrong code into our error correction model.
>
> (2) Thanks for the reminder, the following is another attempt we made in the early stages of the experiment. For the error correction system, another method that does not require unit testing: the user directly inputs the wrong translations of the translation model into all sub_ECDs under $k\text{NN-ECS}_{m}$  at the same time, and then the user verifies whether there are corrected translations in the $m$ outputs .
>
> It's just a matter of which way to run the individual sub_ECDs under the same error correction system is better. It doesn't affect the internals of the individual sub_ECDs, and the final error correction results.
> That is to say, for this method, the error correction performance is the same as the method mentioned in the paper. The detailed data of this method (no unit testing) is shown in Table 3 and Table 4, where the intersection of all sub_ECDs under the same $k\text{NN-ECS}_{m}$ is the error correction result of the system. This process is a bit like a translation model (beam_size = $m$), which outputs $m$ results for the user to verify.
>
> If you also think this method is more effective, we will make modifications later.

---

### Meta-Review · Area_Chair_4uyj · 2023-11-29

**Metareview:**

This paper augments program translation with a retrieval-based component. By anonymizing the retrieved snippets (“unified name rule”) the model significantly improves the performance of baseline methods.

#### Strengths:
- An innovative approach to fixing bugs in translated code.
- Good evaluation and significantly improved results.

#### Weaknesses:
- More rigorous evaluation of the method across multiple languages and project sizes would have been useful.
- Interpretability is not measured via human studies.


Given the importance of the domain and the novel use of the existing method, I recommend this paper to be accepted. I encourage the authors to include all the new experimental results in their paper along with the points clarified in the reviews.

**Justification For Why Not Higher Score:**

While the contribution of this work is interesting, it cleverly applies and existing technique to a narrow domain.

**Justification For Why Not Lower Score:**

Program translation is an interesting research area. Rejecting this work would reduce the visibility of a reasonable method that makes good improvements in performance.

---

### Decision · Program_Chairs · 2024-01-16

Accept (poster)